# Using Artificial Neural Networks for the Estimation of Subsurface Tidal Currents from High-Frequency Radar Surface Current Measurements

Max C. Bradbury * and Daniel C. Conley 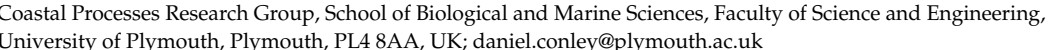

Coastal Processes Research Group, School of Biological and Marine Sciences, Faculty of Science and Engineering, University of Plymouth, Plymouth, PL4 8AA, UK; daniel.conley@plymouth.ac.uk
* Correspondence: max.bradbury@plymouth.ac.uk

**Abstract:** An extensive record of current velocities at all levels in the water column is an indispensable requirement for a tidal resource assessment and is fully necessary for accurate determination of available energy throughout the water column as well as estimating likely energy capture for any particular device. Traditional tidal prediction using the least squares method requires a large number of harmonic parameters calculated from lengthy acoustic Doppler current profiler (ADCP) measurements, while long-term in situ ADCPs have the advantage of measuring the real current but are logistically expensive. This study aims to show how these issues can be overcome with the use of a neural network to predict current velocities throughout the water column, using surface currents measured by a high-frequency radar. Various structured neural networks were trained with the aim of finding the network which could best simulate unseen subsurface current velocities, compared to ADCP data. This study shows that a recurrent neural network, trained by the Bayesian regularisation algorithm, produces current velocities highly correlated with measured values: $r^2$ (0.98), mean absolute error (0.05 ms$^{-1}$), and the Nash–Sutcliffe efficiency (0.98). The method demonstrates its high prediction ability using only 2 weeks of training data to predict subsurface currents up to 6 months in the future, whilst a constant surface current input is available. The resulting current predictions can be used to calculate flow power, with only a 0.4% mean error. The method is shown to be as accurate as harmonic analysis whilst requiring comparatively few input data and outperforms harmonics by identifying non-celestial influences; however, the model remains site specific.

**Keywords:** high-frequency radar; neural networks; tidal resource assessment; ocean currents

## 1. Introduction

As the demand for electricity increases globally, with the concurrent commitment of many countries to lower emission levels, the number of renewable energy developments is soaring. Tidal stream is likely to play a role in this increase due to the predictability of its power, unlike the other offshore technologies of wind and wave. The kinetic energy caused by flood and ebb tides is too low in most areas. However, in some locations, the combination of tidal factors and local bathymetry can result in velocities that have an energy potential that is high enough over a large spatial and temporal range in order to enable production of electricity at a cost-efficient rate, potentially even higher than an efficient wind site [1]. Currently, tidal energy is a maturing technology with multiple single devices deployed and arrays in the planning stages, the most progressed of these being the 398 MW Meygen Tidal Project in Pentland Firth. As high-energy sites are developed, and turbine technology improves to viably produce energy at lower velocities, resource assessments will need to be conducted for site characterisation of new areas.

The rate of movement and directionality of water are caused by influences from tidal harmonics, wind, depth, and other factors, each being specific to a site. Site characterisation is important for tidal energy as current speeds are the primary determining factor for

power, translating to revenue for a developer. In situ measurements are essential for tidal resource assessment, with additional analysis using modelling or harmonics [2]. ADCPs are useful for point measurement resource assessment, especially for measuring turbulence and local variabilities [3,4]. To increase the spatial coverage, Gooch et al. [5] employed spatial interpolation using ADCP data to display the tidal velocity patterns over an area, with the inclusion of the tidal phase difference. Other research towed an underway ADCP around Pentland Firth at a high pace and resulted in the ability to resolve the vertical velocity profiles of the tidal current including its spatial and temporal anomalies [6], this has a high resource consumption and only provides data for a short period. It was shown in their research that the combination of their in situ results with the constraints of a numerical model could produce an accurate four-dimensional representation of tidal velocity outputs. Evidently, observations from ADCPs do show that they are effective to use, especially in the measurement of turbulence and small-scale variations, and also as validation for hydrodynamic models. However, ultimately, they are disadvantaged by their inability to easily assess the spatial and temporal range required in a resource assessment [7]. Alternatively, modelling has proven its potential for current mapping through validation by ADCPs and is now often used for tidal resource assessment [8–10]. However, these require high computational power as well as a lot of detailed data including bathymetry and boundary conditions, which may not be available in many worldwide locations.

Harmonic analysis methods predict the amount of tidal forcing at a point as spectral lines which represent the sum of a set of sinusoids at specific frequencies (cycles per hour). These are obtained as combinations of the totals and differences of integer multiples of six fundamental frequencies, named Doodson Numbers [11], which come about from the motion of celestial bodies [12]. In order to define the amplitude of each frequency, harmonic analysis uses the least squares fit. The amplitude and phase of each frequency characterise a compression of the data in the complete tidal time series. Harmonic analysis is a useful tool for tidal prediction at a point but has a number of drawbacks, namely, the long measurement history required for accurate predictions [13].

The use of shore-based high-frequency (HF) radars for the remote sensing of offshore surface currents and conditions has become increasingly more prevalent [14–16], but has been applied on few occasions for the assessment of subsurface currents. Measurement of surface currents from HF radar works through the transmission of vertically polarised electromagnetic waves which are intercepted and are returned causing an energy spectrum at the receiver. The reflection, when used for ocean currents, is in the form of a Bragg scatter, which results from the reflection of energy by ocean waves with exactly half the wavelength of the transmitted radar waves [17]. Bragg scatter is used because it is the strongest return. The backscatter is returned to the radar carrying information of the surface current velocity and wave spectra. Studies by Thiébaut and Sentchev [18,19] did incorporate a technique using an HF radar, principal component analysis numerical modelling, and a depth power law correlated with ADCP measurements, resulting in a three-dimensional grid of tidal current variability and power density in the water column. It was found that the power available in the bottom layer of their study area was three times lower than near the surface. This is important for the assessment of the optimum hub height of any potential tidal turbine which may be deployed, and the variability of tidal strength allows for design loads for the support structures of devices to be recognised. The HF radar in the study allowed them to apply the technique to the entire area while using real, remotely sensed data rather than modelled, proving the usefulness of the combination of remote sensing with field measurements. New techniques that may increase the ease of assessment are always sought after; an Artificial Neural Network (ANN) could prove a more simple and quicker method than modelling to achieve the same outcome.

ANNs are mathematical models which work similarly to the biological nervous system. ANNs have been extensively used for the prediction of natural processes over the last 30 years, including many successful applications within the marine environment [20,21]. They have shown their worth in tidal range prediction, instead of harmonic analysis,

demonstrating their ability to predict 30 days of hourly tidal height variation using only a small initial dataset and learning period of one day, in contrast to the length of records required by harmonic analysis [22–26].

For tidal analysis, a recurrent neural network (RNN) architecture is preferable, which are capable of learning features and long-term dependencies from time-series data [27], making it an appropriate choice for the oscillatory nature of the tides to get a sense of where the wave amplitudes are likely to be heading. The defining equation of the RNN is such that given values of the time series, $y(t)$, and the input series, $z(t)$, the model is able to predict new values of $y(t)$ [28].

$$y(t) = f\big(y(t-1), y(t-2), \ldots, y(t-n_y), z(t-1), z(t-2), \ldots, z(t-n_z)\big)$$

The $n$ past values are tapped delay lines, storing previous $y(t)$ and $z(t)$ values. The recurrent feature of the network is where these values are regressed onto the new input signal.

The aim of this paper is to assess the capability of a technique combining HF radar surface currents and ANNs for quantification of subsurface currents, to show comparable accuracy to in situ measurements and harmonic analysis, decreasing the resources required for a reliable tidal stream resource assessment of a large area. This will be achieved through the creation of various structured neural networks to find the highest performing network for subsurface current prediction. The ANN will be validated through statistical comparison to an independent ADCP dataset, and subsequently used for tidal power calculations. The ANN will then be used to associate HF radar surface currents to subsurface currents at another location, followed by discussion of network capabilities and behaviours.

## 2. Materials and Methods

### 2.1. Site and Datasets

The Celtic Sea off the north coast of Cornwall has a high-energy wave regime, suitable for the deployment of wave energy converters. The site is not a potential candidate for a large tidal stream development but is suitable for demonstrating this technique. The tidal movement is predominantly meridional, with a lesser zonal component [29], due to the proximity and morphology of the coast.

Data for the ANN were pre-collected (Conley, 2013, unpublished data), continuously available between March and December 2012.

The surface velocity data used as inputs for the ANN were obtained from a system of two high-frequency Wellen Radar stations positioned 40 km apart at Pendeen and Perranporth and overlooking the WaveHub test site on the north coast of Cornwall (Figure 1). At each site, there is a 16-element phased-array receiver and a square four-element transmitter orientated parallel to the coast. At Pendeen, the receiver is orientated 113° clockwise from true north, hence its boresight is directed 23° from north. At Perranporth, the receiver is orientated 35° from north with its centroid beam pointed 305°N so it aligns with the prevailing westerly swell. The radar stations independently measure the surface current velocity for 17 m 45 s every hour with a range resolution of approximately 1 km and angular resolution of 7°. The radars use a "listen before talk" mode [30], which determines the best frequency for transmission within a 250 kHz bandwidth centred on a frequency of 12 MHz. This results in transmitted waves being backscattered off ocean waves 12.5 m long at a range of up to 101 km. Surface currents are recordable over the full range while wave products are only available over half the range due to the large signal-to-noise ratio of the second-order returns compared to the first-order echo. The backscattered information is transformed into an orthogonal coordinate system to set it to a 1 km grid [16].

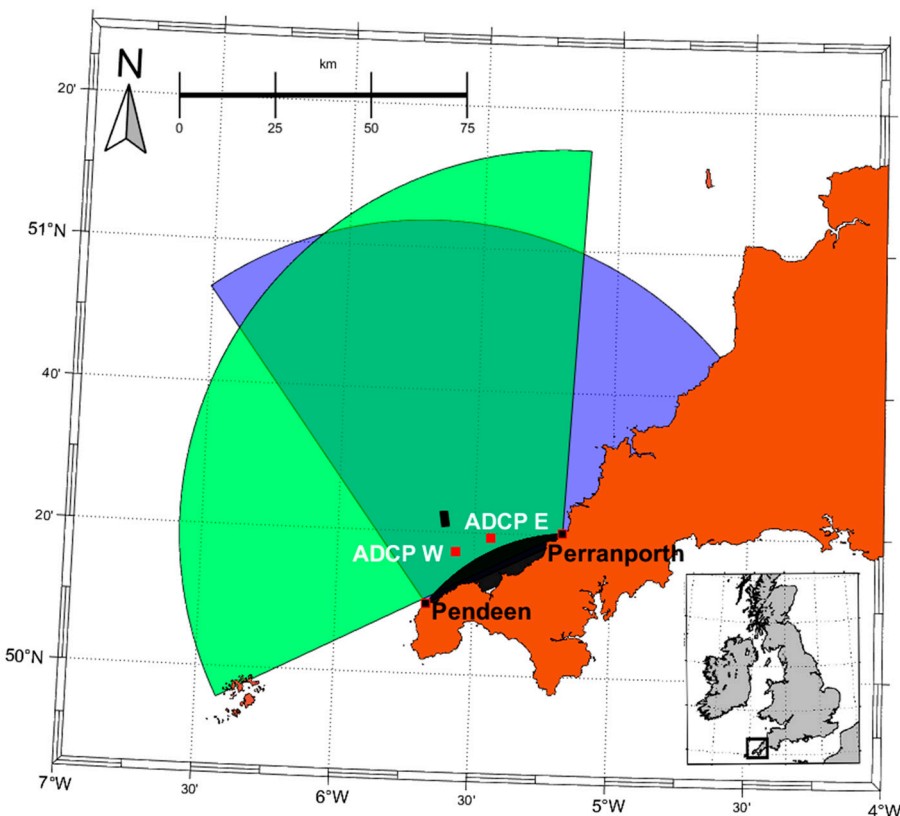

**Figure 1.** Map showing the locations of the HF radars at Pendeen and Perranporth and their coverage. The red squares represent ADCP west and east. The rectangle represents the WaveHub test site.

Two upward-looking Teledyne RD WorkHorse ADCPs were deployed to collect subsurface velocity data to train and validate the ANN. ADCP-West was placed 16 km from Pendeen and 29 km from Perranporth, deployed at a mean depth of 34 m, while ADCP-East was located 24 km from Pendeen and 19 km from Perranporth deployed at a mean depth of 37 m, with a separation of approximately 10 km between the two. The ADCPs operate at 600 kHz in the Janus configuration with four beams located 20° from vertical. Current velocities were measured at bin depth intervals of 0.75 m every 10 minutes at 2 Hz, with the first bin being 1.86 m above the seabed. At 600 kHz, the accuracy of the sensor is ±0.3%. The ADCPs were periodically recovered for data retrieval and battery replacement, then redeployed as close to the previous location as possible.

### 2.2. Metrics Used in Neural Network

The metrics used as inputs to the ANN in this paper must be covered by the radar or be from readily available data to maintain the advantage of this method over traditional methods. This means that while density differences and other factors may have had some influence on the subsurface currents, they were excluded. The metrics used as inputs are: the surface velocity above where the subsurface current was to be predicted, along with the surrounding four surface velocities and the tide varying depth. The surrounding eight velocities were also trialled. However, the network had lower performance. Wind velocity and wave field data could have been obtained and applied to the network. However, Lu and Lueck found that 91% of the subsurface flow velocity in their test site could be attributed to the lunar- and solar-influenced tides [31], showing these additional methods would be of small importance while adding extra neurons and training time to the network. Additionally, the network performed sufficiently well without these inputs, so they were not added to reduce network complexity. This ANN technique combined with the radar has a huge advantage over an ANN alone, the constant radar input means that errors will

increase less over time, whereas an ANN alone would soon begin predicting off its own predictions and reduce in accuracy over time.

Both the data processing and ANN creation were carried out in MathWorks' Matlab and the Matlab Neural Network Toolbox.

The surface velocities above the ADCP which were to be used as inputs to the ANN were identified using the average coordinates of the ADCP placement locations. Surface velocity time series were made at these locations. Two linear interpolations were applied to the data; first of all, the HF radars occasionally failed to record if the data received from a particular cell were below a quality threshold, these were interpolated using their surrounding time points to provide continuous data. Secondly, due to the difference in sampling rates and times of the two instruments, the ADCP data, which collected data more frequently, was linearly interpolated over the times at which the surface velocities were collected, producing surface, subsurface velocity (both east and west components), depth and time data with 6825 hourly time points spanning the same period. The upper 10% of the water column was removed from the ADCP data as this near-boundary region is subject to sidelobe interference [32]. The radar data were then arranged into a format which the ANN would take as an input, while the ADCP data would be the output.

*2.3. Neural Network Creation and Analysis*

The architecture of the RNN in this work consists of 6 input neurons, equivalent to the five radar surface velocities surrounding the ADCP and one depth predictor. The predicted velocity at 52 depth bins is represented by the network output, thus, there are 52 neurons in the output layer. The learning ability of an ANN is dependent on the architecture. If the network is too small (too few hidden neurons), it may not have a large enough degree of freedom to learn the relationships between the data. Whereas, if the number of hidden neurons is too large, it can bring about overfitting, where the network fails to generalise with new datasets. The number of hidden neurons was varied (1–50), and the highest performing was chosen, based on statistical tests explained further down this section, comparing unseen data and the network's predictions.

Several training functions were also employed to obtain a network with the highest performance and generalisation to new data. These training methods were, the gradient descent method with adaptive learning rate (GDA), the scaled conjugate gradient method (SCG), the Levenberg–Marquardt algorithm (LM), and the Bayesian regularisation backpropagation method (BR), which is based on LM.

In formulating the network, available data for 2 weeks were used, representing an entire tidal cycle, making 336 hourly time points (6–19 June 2012). Data for more than 2 weeks were also used for training (4 and 8 weeks), but there was no improvement in performance. Performance began to deteriorate once data for 1 week were used. These training data were further split into 70% for training, 15% for model validation, and 15% for testing. The data were divided sequentially, rather than randomly, in order to enable the feedback delays in the recurrent to learn relationships between neighbouring data points. This left any of the remaining 6151 time points for manual testing to assess the capability of the ANN on predicting unseen data. The model was trained by the reduction in the mean square error (MSE) criterion to evaluate performance during training.

Since the performance of a network varies between each training session with the same inputs, due to its ability to find different solutions to problems, each network was required to be trained multiple times in order to obtain a high-performing network. Following the calculations of Iyer and Rhinehart [33], the networks should be trained 90 times each to be 99% confident that the best version trained was within the best 5% of possible networks. Through undertaking network training in a loop, starting with random initial weights each time, the 4 training functions, along with trialling 1 to 50 hidden neurons, trained 90 times each resulted in 18,000 networks being trained. These were assessed using the statistical tests below. Tapped delay lines were placed connecting the output of the first to fourth hidden neurons back to the input of the first neuron to let the network have

memory of the previous four timesteps to predict the next. This number of lines produced the highest performing networks; when there were less than four hidden neurons, delay lines were present on all hidden neurons. Once trained sufficiently, the network was used to predict velocities at any height above the seabed, at any time in the available data, and subsequently used to produce products necessary for tidal resource assessment [2]. In addition, the networks were used to predict subsurface currents at the location of ADCP-E, 10 km from where the network was trained.

Statistical tests were used to assess the network's performance on unseen data. It is imperative to use multiple statistical tests as single tests such as the coefficient of correlation might show good correlation for consistent errors. The tests used were: coefficient of correlation (r), coefficient of determination (r$^2$), root mean square error (RMSE), mean error (ME) (to show prediction bias), mean absolute error (MAE), mean absolute percentage error (MAPE), and the Nash–Sutcliffe hydrological efficiency (used to assess the predictive power of hydrological models where the output ranges from $-\infty$ to 1, and where E = 1 would be a perfect match) [34].

## 3. Results

### 3.1. Performance of Neural Network Structures

The independent data on which the network was tested were for 110 days spanning from August to December. The best-performing network size of each training function from the 18,000 created is shown in Table 1, for the east velocity component, along with statistical differences between the network predicted time series and the unseen measured time series. Training speed was between 15 seconds and 2 minutes for all GDA and SCG network sizes, while LM and BR began at <10 seconds, but exceeded two minutes after three hidden neurons were added and training with these functions became impractically slow at 10 neurons so was discontinued. The highest performing network from each training function was similarly capable of prediction of the subsurface currents. The RMSE was sufficiently lower than the standard deviations of the models, showing good predictability. The best-performing network was BR with 1 hidden neuron, shown by its high r, r$^2$, and E values, along with its notably lower error values. In addition, the BR model was carried forward due to its reduced complexity whilst having the best capability, along with its smoother velocity profile predictions shown in 3.2. Results for the north component ANN showed slightly inferior performance, but still very high. BR with one hidden neuron was also the best architecture for the north component (Table 2).

**Table 1.** Best-performing east network for each training function with their respective statistics, rounded to three decimal places.

| Network Function | Hidden Layers | r | r$^2$ | STD (ms$^{-1}$) | RMSE (ms$^{-1}$) | ME (ms$^{-1}$) | MAE (ms$^{-1}$) | MAPE (%) | E |
|---|---|---|---|---|---|---|---|---|---|
| GDA | 22 | 0.986 | 0.972 | 0.514 | 0.082 | 0.0114 | 0.060 | 9.915 | 0.971 |
| SCG | 27 | 0.987 | 0.974 | 0.511 | 0.081 | 0.0036 | 0.054 | 1.372 | 0.974 |
| LM | 1 | 0.988 | 0.976 | 0.511 | 0.081 | 0.0037 | 0.056 | 2.757 | 0.975 |
| BR | 1 | 0.989 | 0.978 | 0.464 | 0.067 | 0.002 | 0.048 | 4.630 | 0.979 |

**Table 2.** Best-performing network for north component with statistics. MAPE = N/A due to zeros in data, rounded to three decimal places.

| Network Function | Hidden Layers | r | r$^2$ | STD (ms$^{-1}$) | RMSE (ms$^{-1}$) | ME (ms$^{-1}$) | MAE (ms$^{-1}$) | MAPE (%) | E |
|---|---|---|---|---|---|---|---|---|---|
| GDA | 22 | 0.961 | 0.924 | 0.247 | 0.127 | −0.018 | 0.059 | N/A | 0.917 |
| SCG | 27 | 0.975 | 0.951 | 0.251 | 0.050 | −0.008 | 0.046 | N/A | 0.949 |
| LM | 1 | 0.982 | 0.964 | 0.263 | 0.043 | −0.008 | 0.039 | N/A | 0.963 |
| BR | 1 | 0.979 | 0.958 | 0.239 | 0.050 | −0.005 | 0.0037 | N/A | 0.958 |

The scatter plot in Figure 2, showing predicted vs. measured velocities over the time series using the best BR network, shows that the network can reliably predict tidal velocities from the short training dataset (2 weeks).

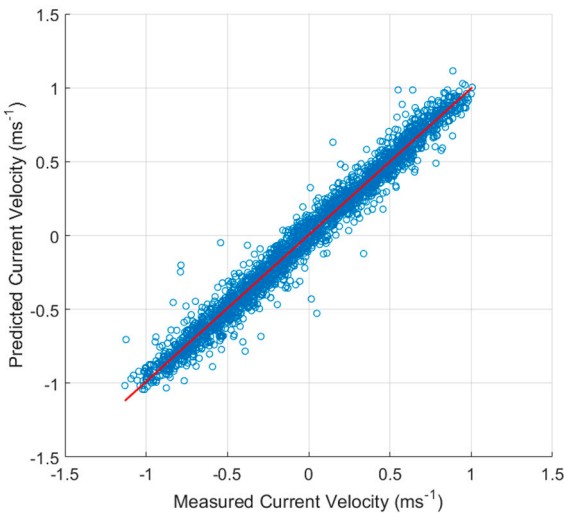

**Figure 2.** Scatterplot of the measured vs. predicted east velocities at 20 m above the seabed by BR, around the exact fit line.

### 3.2. Vertical Velocity Profile

Knowledge of the vertical variation in current velocity is a critical step in tidal resource assessment and was also important in this work to confirm BR as the highest performing network. While it seemed the GDA and SCG functions were capable of accurate prediction of current time series at select depths, upon averaging the vertical current profile on 16 spring flood tides, these methods predicted a large net underprediction. Through visual assessment and by using r and ME, the BR function produces the profile which best represents the real measured profile (Figure 3, Table 3). The depth-averaged velocity predicted by the network was 0.833 ms$^{-1}$, compared to 0.824 ms$^{-1}$ measured. r and ME could not be calculated above 29 m where the ADCP data had been removed.

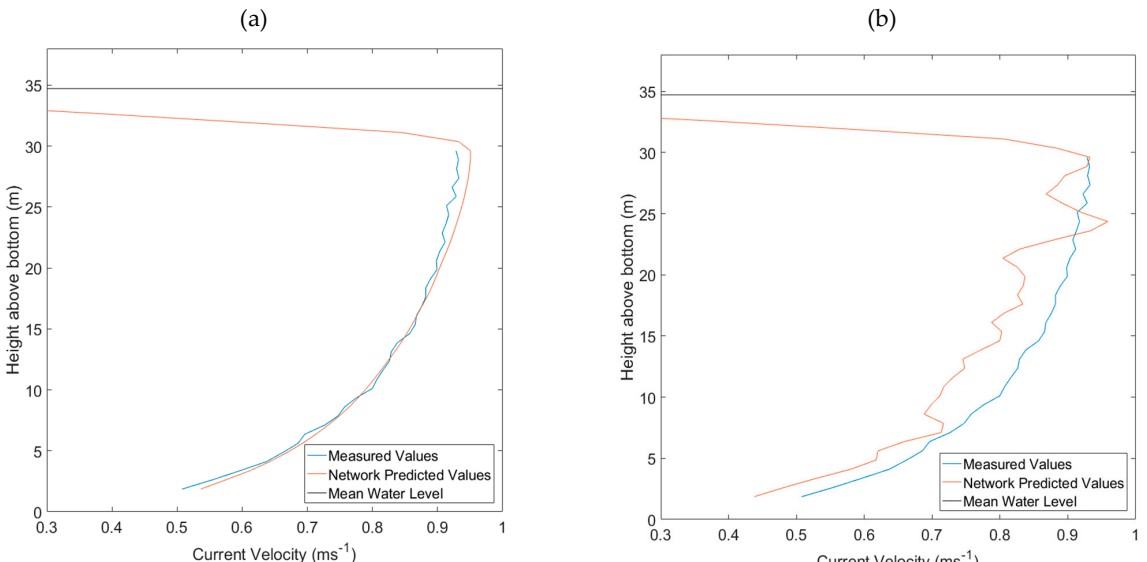

**Figure 3.** (**a**) Current velocity profile using the BR training function. (**b**) Current velocity profile using the GDA function. Blue: measured values, orange: network prediction, and black: mean water level.

**Table 3.** r and ME of the different trained functions on the predictions of vertical velocity profiles.

| Function | GDA | SCG | LM | BR |
|---|---|---|---|---|
| r | 0.925 | 0.989 | 0.996 | 0.997 |
| ME | −0.0519 | −0.0170 | 0.0155 | 0.0087 |

BR was also the best training function for the north current profile. These networks were used from henceforth.

### 3.3. Time Series Prediction

Figures 4 and 5 illustrate how the network was able to predict the E-W velocities at all depths of the water column, showing the diminishing velocities toward the seabed, using only surface velocities. The error in prediction shown in Figure 4b is mostly contained below ±0.05 other than a few exceptions. The measured and predicted velocities are in phase and the fortnightly cycle is reproduced.

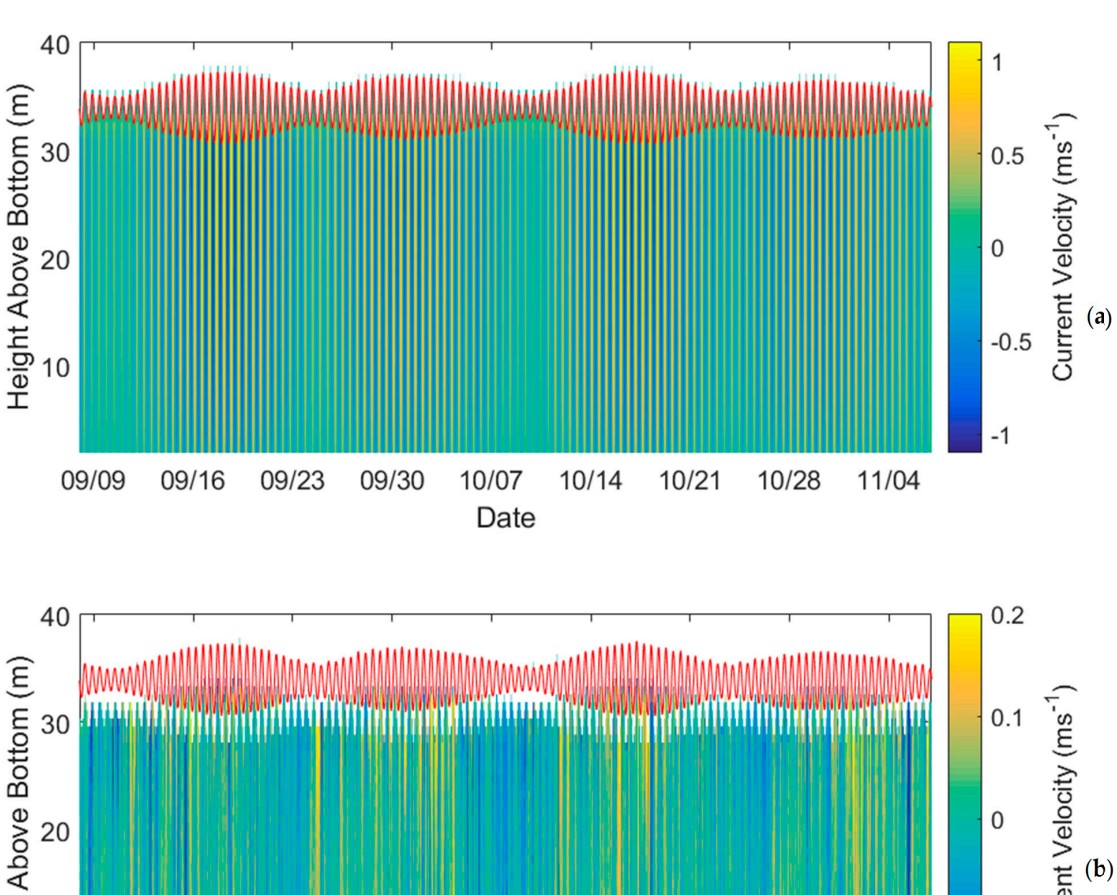

**Figure 4.** (**a**) Plot of the predicted current velocity throughout the water column. (**b**) Error between the predictions and measured data. Red line = water depth. Top 10% is white despite the ANNs prediction as the data were removed in the ADCP comparison due to sidelobe interference.

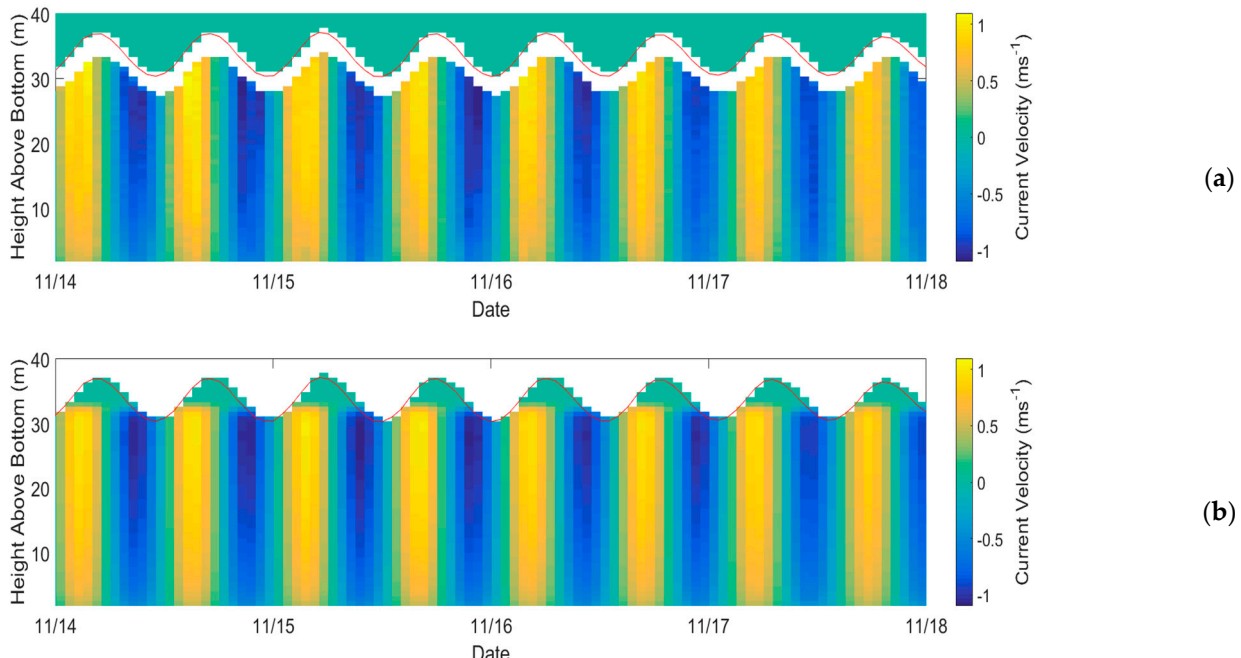

**Figure 5.** (**a**) Enhanced comparison of measured velocity variation. (**b**) Network predicted velocity variation. Red line = water depth. White space below red line is the location of the inaccurate ADCP data.

The predictions for the north component network also showed good agreement, although the error figure contained more instances of blue, showing a general underprediction, more so at neap tides.

### 3.4. Total Velocity and Direction

Figure 6 shows the current rose generated from measured and network predicted values, once the northern and eastern current velocities had been combined, showing good predictability of the network.

### 3.5. Tidal Power

The tidal power was calculated using the combined velocities. The impact of the incident current angle on the power take-off of a non-yawing bi-directional turbine was calculated by adding the cosine response. Table 4 shows the measured mean raw power before the angle was considered, followed by the measured power considering the angle, and then each network predicted power considering the angle, showing again that BR predicted the best network, closely followed by LM.

Figure 7 shows a short period of measured, and network predicted powers. The network appeared to underpredict some of the largest, abnormal peaks but overpredicted the medium-sized peaks. This pattern of predictions resulted in a mean power difference of only $0.51\,\mathrm{Wm^{-2}}$ over the 3.5 months. The inclusion of the cosine response showed that the tides have a very high angular fidelity, showing only $0.58\,\mathrm{Wm^{-2}}$ decrease in power. The power–frequency plot (Figure 8) shows high similarity between network and measured values, showing that using the network predicted or measured values may have little impact on power prediction.

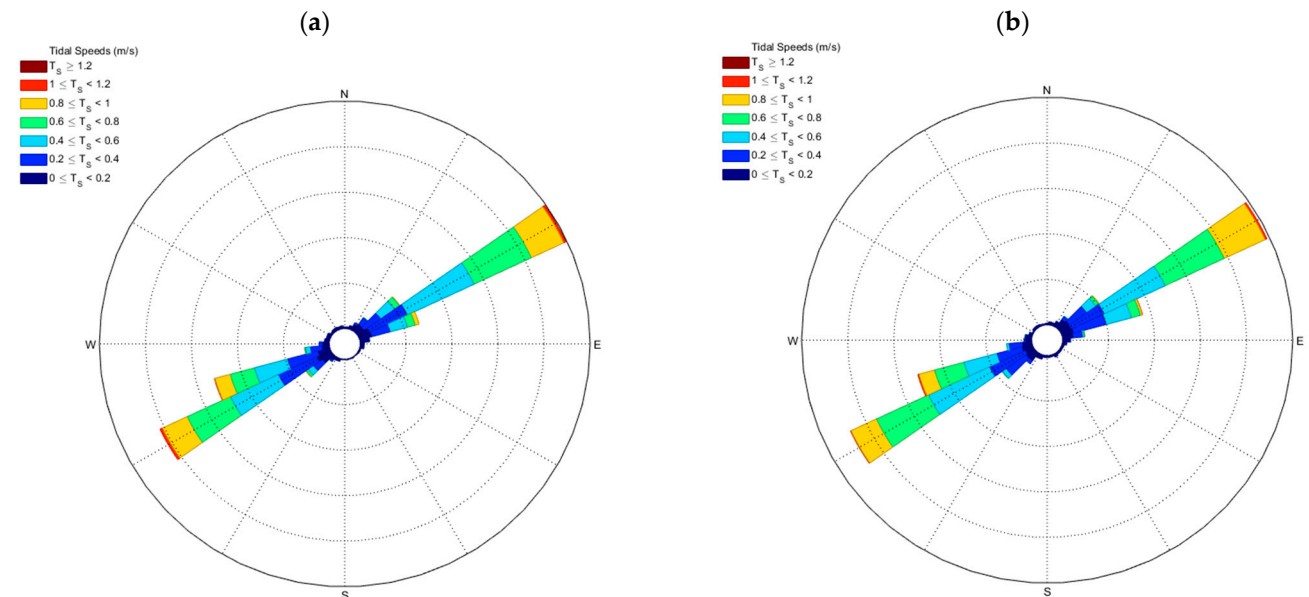

**Figure 6.** (**a**) ADCP measured current rose. (**b**) Network predicted current rose.

**Table 4.** Measured mean power before accounting for the angle of current (raw power), considering incident angle (measured), and each training method's best-performing networks prediction of mean power and error.

|  | Mean Power (Wm$^{-2}$) | Network Percentage Error from Mean Measured (%) | Max Power (Wm$^{-2}$) |
|---|---|---|---|
| Raw power | 127.72 | - | 904.95 |
| Measured | 127.14 | - | 904.82 |
| GDA | 124.03 | $-2.48$ | 928.03 |
| SCG | 126.17 | $-0.77$ | 777.68 |
| LM | 128.10 | 0.75 | 931.42 |
| BR | 126.63 | $-0.40$ | 875.88 |

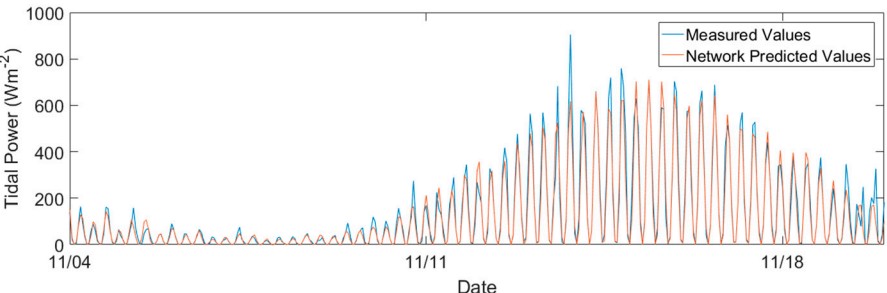

**Figure 7.** Extract of time series of measured and predicted current power. Blue: measured values, and orange: network prediction.

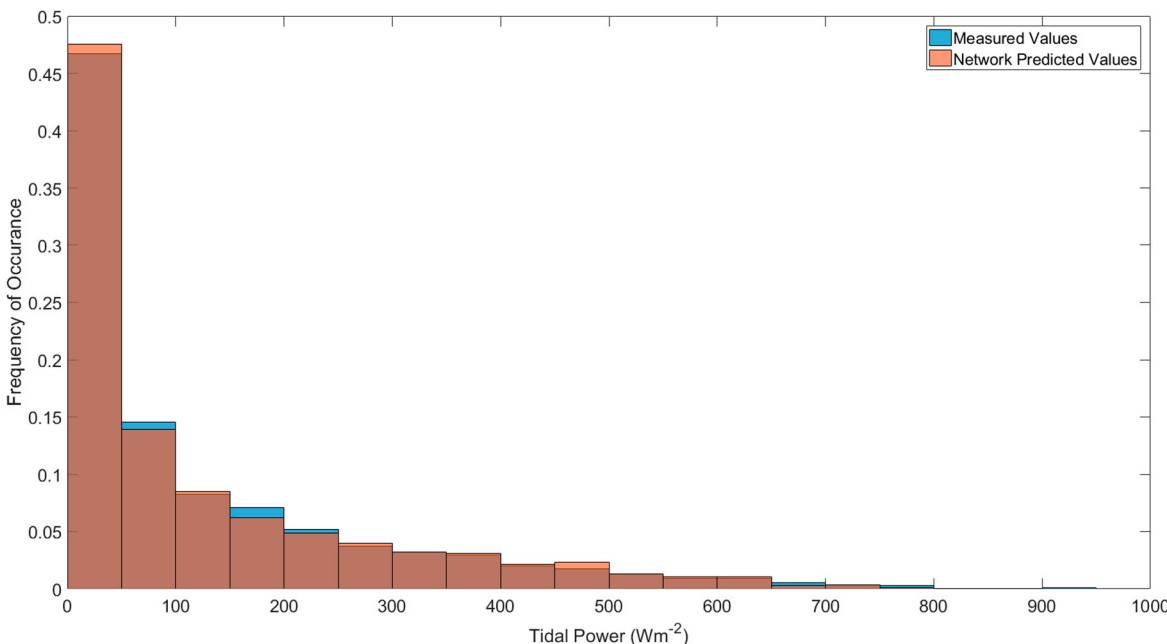

**Figure 8.** ADCP measured and neural network predicted frequency of tidal power occurrence separated into 50 Wm$^{-2}$ bins. Blue: measured values, and orange: network prediction.

### 3.6. Application to Other Areas

Using the trained network to predict nearby subsurface currents would be a pinnacle finding to reduce the resources required for a resource assessment. The current velocities at the east ADCP were consistently slower than at the training location. Using the same network as the previous sections, the network consistently overestimated peak velocities. Despite the mean absolute velocity difference between the measured and predictions being only 0.03 m s$^{-1}$, this translated to a large difference in the resulting mean power; 29.73 Wm$^{-2}$ measured and 41.25 Wm$^{-2}$ predicted.

To achieve better predictions at ADCP-E, with a network trained at ADCP-W, a number of network modifications were made. Firstly, the SCG network with the optimum 27 hidden neurons produced the most accurate predictions, along with increasing the number of delay lines to 12. Finally, removing the depth as input improved predictions at ADCP-E, due to the different bathymetry. Training of the ANN using the same period of data it was to be tested on also improves predictions, i.e., when trained using August-October at ADCP-W, it could better predict the subsurface currents 10 km east over the same period. This greatly lessened the consistent overprediction, reducing the mean current difference to 0.016 m s$^{-1}$ and reducing mean power difference to 4.8 Wm$^{-2}$ (36.83 Wm$^{-2}$ measured and 41.59 Wm$^{-2}$ predicted), peaks were also suppressed, although still higher than the ADCP-W predictions, shown by the far higher MAPE (Table 5).

**Table 5.** Statistics of the ADCP-E prediction by SCG network.

| Network Function | Hidden Layers | r | r$^2$ | STD (ms$^{-1}$) | RMSE (ms$^{-1}$) | ME (ms$^{-1}$) | MAE (ms$^{-1}$) | MAPE (%) | E |
|---|---|---|---|---|---|---|---|---|---|
| SCG | 27 | 0.976 | 0.953 | 0.377 | 0.079 | −0.013 | 0.054 | 10.68 | 0.941 |

Comparison of the mean spring flood velocity profiles in Figure 9 shows that at ADCP-E, there is a large overprediction bias in all but the lowest 6 m of the water column. The largest error being 0.1 ms$^{-1}$ (14.60% error). The network similarly overpredicts velocities during ebb, contributing to the overall negative mean error of the network.

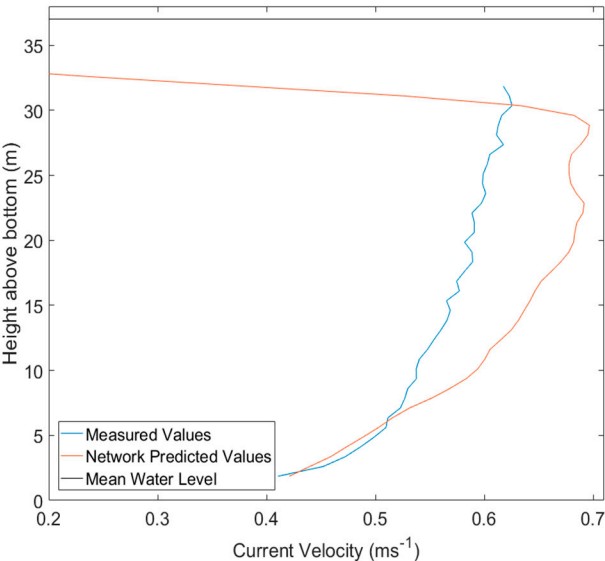

**Figure 9.** Velocity profile at ADCP-E, trained by SCG method using ADCP-W. Blue: measured values, and orange: network prediction.

## 4. Discussion

### *4.1. Neural Network Performance and Behavior-ADCP-W*

#### 4.1.1. Current Velocity Time Series

Despite all training functions being able to adequately predict both the pattern and magnitude of the tidal velocities, it was imperative to choose the best-performing network as cubing the velocity for power would enhance inaccuracies. The RMSE values ranged from between 13–16% of their corresponding STDs while the Nash–Sutcliffe Efficiency was always over 90%, suggesting high prediction efficiency [35]. While the difference in errors is minimal between training function outputs due to all functions accurately predicting the majority of the tidal cycle, the small difference in error was caused by the network's differing abilities to predict peak currents. The GDA function was the most variable, often underestimating during spring tides, sometimes by 0.1 ms$^{-1}$ at peak ebbing tide. As found previously in a wide variety of problems, the LM functions, including BR, which is based on LM, outperform the simple gradient descent and scaled conjugate gradient methods [36–39]. Based upon r and MAPE values of the BR network (0.99 and 4.63%, respectively), this model can be described as a good predictor for tidal modelling [40]. The high-quality statistics of the BR network in comparison to the measurements show that the HF radar-ANN technique proves a useful tool for analysis of subsurface current time series over any period at the training location.

The major limitation of the ANN is that it is a black box model, failing to simulate the internal physical processes of a tidal system. The simulation of this is of vital importance for resource assessments. In addition, because the black box model does not allow insight into the calculations made to reach the target, assumptions of hydrodynamics which caused network behaviours are made.

Analysis of the colour plots (Figures 4 and 5) shows that the network generally has a positive bias around spring tide, and negative around neap tide. To determine the cause, a network was trained over a different period to identify if unique conditions during the initial two weeks caused errors. There was no significant difference in errors to the original training period. The error must, therefore, be in the network's application of weights and calculations.

Along with the above small-magnitude long-period bias, some short but pronounced errors also exist. Where they occur, they are present throughout the entire water column, identifiable as distinct bars, a different colour to their surroundings. The network predicts the time series of each depth bin independently [41], not considering the relationship

between the present bin and those above and below. Therefore, as the network learns the recurring features of each depth time series, it is unlikely to make the same error independently at each bin. The resulting implication is that errors between predictions and the measured values are caused by abnormal surface currents during training, causing the network to predict accordingly, or abnormal ADCP measurements in the testing data.

Several behaviours of the network were noted throughout the analysis. The network attempted to predict the upper 10% of the water column (Figures 4a and 5b), even though the data were removed from its training data. It is impossible to confirm if the network's predictions for this region are correct. However, some suspicions may be discussed. The upper 10% looks to be well predicted at low tides due to the continuation of the lower currents to the surface, whereas at high tide, the velocities unrealistically approach $0 \text{ ms}^{-1}$. It appears the network tries to draw from any values in a depth bin, so as the low waters are more often submerged and below the 10% threshold, there is more for the ANN to learn from. At depth bins always in air or the threshold, the network can only predict $0 \text{ ms}^{-1}$, and heights which are mostly in the upper 10% are inaccurate due to limited data.

### 4.1.2. Velocity Profiles

In the prediction of the velocity profiles, the alike functions behaved similarly, BR and LM net-overpredicted while GDA and SCG underpredicted. The BR function predicted the closest fit to measured values (99.69%), as well as the most natural-looking decrease in velocity with depth. The GDA method produced the lowest correlation (92.49%) with a much more sporadic pattern than other methods. This is likely because the appropriate learning rate could not be found, despite running numerous networks with various learning rates, in order to find the minimum error and eventually incorporating an adaptive learning rate. When the learning rate is too large it will not reach the target as it always overshoots, if too small, the network may never have reached the target [42,43]. This likely occurred differently at each depth bin due to the random initial weights chosen and various routes the network takes to find the answer. The LM methods made better predictions. This is because they interpolate between the gradient descent method and the Gauss–Newton method [44], using the latter more when the parameters are close to the target, reducing the square error by assuming the least squared function is locally quadratic, then finding the minimum of the quadratic. This results in less overshooting. The BR method uses the LM method but minimises a combination of the square errors and weights, then determines the correct combination to produce a better generalising network [45], hence the best predictions on the test data.

Using the BR function, it is obvious from the velocity profile that there is a greater overprediction bias at 22–29 m above the bottom (ME = $0.0168 \text{ ms}^{-1}$) than from the seabed to 22 m (ME = $0.0052 \text{ ms}^{-1}$) (Figure 3a). The likely reason for this is the occurrence of a weather system approaching from the southwest in the first half of the two-week training period [46], the south-westerly winds (with some gusts up to $100 \text{ km h}^{-1}$) would have accelerated the surface currents during flood tide and reduced the surface velocity during ebb tide, while as the distance from the surface increases, the effects from the wind diminish. The network learns this to be normal. This hypothesis can be complemented by plotting the mean spring ebb current profile predicted by the network (Figure 10). The underprediction of the surface currents nearest to the surface suggests that the abnormally strong south-westerly wind could be slowing the ebb surface current.

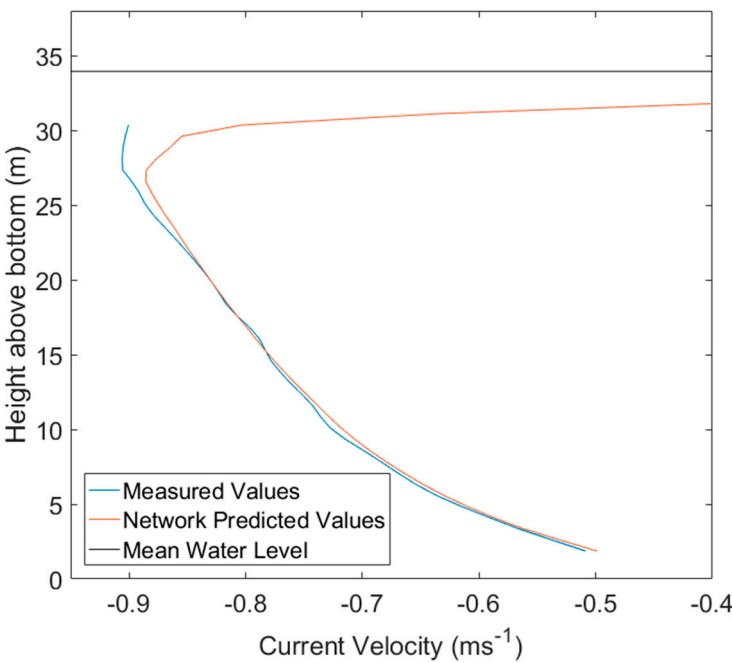

**Figure 10.** Velocity profile at ADCP-W averaged over 16 spring ebb tides using the BR network. Blue: measured values, and orange: network prediction.

### 4.1.3. Flow Power

The statistics and power–frequency graph suggest that at this location, the HF radar-ANN method could be used to reliably estimate power. Misalignment of a turbine with the incoming current direction can reduce power by 6% if the turbine is 20° off the incoming flow direction and 23% if 40° off axis [2,47,48]. This test site is highly bi-directional with the majority of currents being within 5±, applying the cosine response decreases power by only 0.005%, with EMEC stating this will capture 100% of possible power [2]. The currents outside of the 5± are likely below a turbines cut-in speed and are made negligible by the cubing of the velocity for power.

### 4.1.4. Potential of the Radar-Network Technique at a Single Location

All products recommended by EMEC for the use of a model to compliment field surveys have been completed with high agreement [2], proving the potential to replace some long-term in situ surveys. It could also replace models where only the output is required, requiring far less computing power and less cumbersome due to the requirement of models for various data including long time series and bed roughness, whereas the ANN need not understand these physical aspects. On top of this, not all countries looking to develop tidal power have satisfactory record keeping for models. The minimal data for this network would be inexpensive and easy to gather, with 99% accuracy.

The combination of the remote sensing and ANNs outperforms the forecasting ability of models or ANNs alone, as the errors will not increase over time due to the constant radar input. ANNs alone for forecasting hydrodynamic data perform well with high r values (0.90+), but lower Nash–Sutcliffe efficiencies (0.7–0.9) [49], due to the error increase with time since the training data. Despite this advantage of the constant input of the HF radar, Tang et al. showed the prediction of a time series by an ANN is better if the training data gathered are in the short-term history of the testing period [50]. This could mean that errors may begin to appear in the network when long-length tidal harmonics or the modulation of the perihelion occur, for which the network was unable to be trained over. Changes to bathymetry in the vicinity could also vary tidal dynamics [51]. As well as this, the network should be statistically tested seasonally to evaluate performance in non-familiar conditions as this has been shown to vary a network's performance [49]. It is possible this change over

time has occurred in the testing data, errors did seem to increase slightly over time. This is confirmed by the greater MAE of 0.069 ms$^{-1}$ in the last month in comparison to 0.056 ms$^{-1}$ in the first two and half months. This higher-than-average MAE could be due to the long time since the training period, where long-period harmonics may have altered the tide, or the approaching winter could bring about different coastal conditions to the summer training period.

### 4.2. ADCP-E and Beyond

In order to improve the network's predictions at ADCP-E, the aforementioned alterations to the network were required, of which 90 were trained to find a network in the best 5%. The justifications for the alterations were: the reduction in iterations was to prevent the network from overfitting to the specific conditions at the training site. Secondly, the SCG network with 27 hidden neurons allowed the network to learn more complex relationships between the surface and subsurface currents while the increased layer feedback delays to 12 also allowed the network to account for more dependencies which the past current would have on the present. Removal of the depth input improved results, this was suspected to be caused by the original network placing too much weight on the depth input, then, once applied to the east location which is 3 m deeper, it would associate this increase in depth with spring tide and, therefore, higher velocities, overestimating the results.

As the network currently stands, and as acknowledged by research creating networks for tidal range prediction, the model developed is site specific [26]. Although the model in the current work does show good promise at predicting the small magnitude tidal velocities and power at ADCP-E based on time series and statistical tests shown in Table 5, the comparison of velocity profile (Figure 9) proves that these are deceiving and from the middle of the water column upward, there is up to a 15% overprediction bias by the network. Chang and Lin [52] demonstrate an ANN certainly is sufficiently capable of predicting tide heights at sites 10–20 km away from where it was trained ($r^2$ = 0.84–0.95) unless complex bathymetrical variation occurs. This is precisely where the ANN falters on ADCP-E. The difference in morphology between sites is such that at ADCP-W, the mean spring tide measured by the radar at the surface above the ADCP is 0.877 ms$^{-1}$, whereas the highest-depth bin from the ADCP where there is accurate data have a higher mean velocity at spring tide of 0.928 ms$^{-1}$. Therefore, the network has learned to predict higher velocities below the surface than the surface data given as the input. In contrast, at ADCP-E, the mean spring surface current measured by the radar is 0.651 ms$^{-1}$ whilst the ADCP bin highest in the water column has a mean spring velocity of 0.582 ms$^{-1}$, lower than the surface velocity. The network has not seen this relationship before and acts as it did at ADCP-W, causing a large overprediction at ADCP-E. Naturally, to test if the error was due to external factors, the network method was reversed, being trained using ADCP-E data and tested on ADCP-W, the network underpredicted ADCP-W currents by a similar bias. Inter-site bathymetric and coastal morphology differences are why almost all tidal range ANN research has concluded the network can only be used for single location estimation [22–26], while research which has attempted multi-point prediction uses astronomical data as network inputs and concludes that prediction error increases with distance and dissimilarity from the training location [52].

The network structure itself is also causing errors at the new site; during training the network only learnt how to produce 52 height bins of data, so it can only do the same at ADCP-E which is 3 m deeper. The upward-looking ADCP, therefore, expects the bins to be closer to the surface than in reality. This is apparent near the surface, where the ADCP measurements continue for 3 m extra while the prediction makes a sharp decrease toward 0 ms$^{-1}$ because it is trained on data where this depth is within the 10% threshold. The error is lowest near the seabed, as water–seabed interactions will be similar at each site but increase with height where differing surface interactions also play an increasing role.

Due to issues such as the above, a combination of ANN with hydrodynamic model nodes could prove useful and is often used to take advantage of the ANN advantages [53,54].

As ANNs use far less computation power than models, a series of nodes could be created by a coarse grid model, replacing ADCPs, enabling the network to be trained from multiple points and predict the surrounding currents. Alternatively, the method used by Makarynskyy et al. for suspended sediment prediction could be used [54], where a model was used for a short initial period, on which the ANN was trained. In this case, the network would associate modelled subsurface currents with HF radar surface currents, after which, just the radar input would be required, saving huge computation cost.

### 4.3. Comparison to Harmonic Analysis

While the ANN provides a simple way of predicting semi-diurnal tidal velocities at a site without determining the harmonic parameters, non-tidal induced off-nominal currents during training would cause the network to apply assumptions made from these to the test data where the same currents may not occur, while unique currents during testing will not be acknowledged. In comparison, harmonic analysis does not acknowledge any non-tidal currents. A total of 38 constituents were considered including the shallow water M10 constituent, and the analysis was carried out on each vector component separately by least squares fitting using the T_tide Matlab toolbox as described by Pawlowicz et al. [55]. In Figure 11, the extracted harmonic constituents were used to plot tidal velocities, in comparison to the ADCP, to which the network was also compared. The residual signal between the two was also plotted, showing the difference between the harmonic prediction and the real measurements.

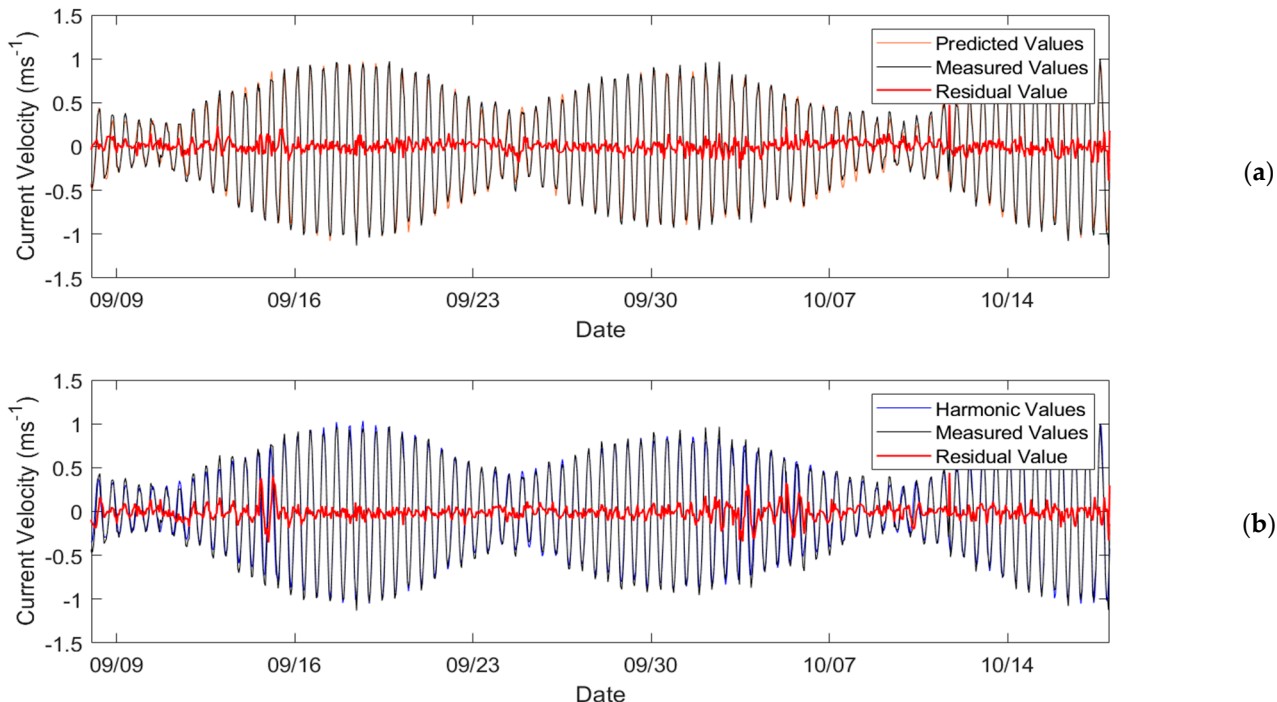

**Figure 11.** (**a**) Network predicted current, alongside harmonic prediction and residual current between the two. (**b**) Measured tides, alongside harmonic prediction, and residual tide between the two. Blue: harmonic prediction, orange: network prediction, black: measured values, and red: residual value.

It can be seen that the residual between the ANN and measurements has fewer large peaks compared to harmonic analysis since the network can consider some small, local velocity fluctuations not caused by the tidal constituents, causing fewer errors. This enhanced accuracy over harmonic analysis is for two reasons. Firstly, the ANN may have learned regular, non-celestial currents from the training data and applied them to the testing data, which harmonics cannot identify. Secondly, the constant radar surface measurements are an advantage in this method, even if there are abnormal currents which do not strictly

follow the tidal wave, the network has learned how to associate these with the subsurface currents. An example of this is at approximately 9/15—the tidal cycle seems to occur an hour later than the harmonics would suggest, so there is a large residual, but the lack of error in the network values shows that the network could predict this deviation from celestial tides. However, the analysis also shows that the network sometimes ignores large, multi-hour, non-tidal variations such as the large error spike at approximately 10/11. There is a large error in both the network and harmonic predicted values due to a seemingly random decrease in current velocity (the anomaly does not coincide with an ADCP retrieval). The sudden change in current velocity was not present in the radar input data so it existed only in the measured subsurface currents and, therefore, impossible for the network to predict; perhaps too sudden for the recurrent delay lines of the network to anticipate. Comparison of the plots complements the network's capability in predicting a portion of the real amplitude of current peaks which harmonic analysis overlooks, as long as the abnormal current is present on the surface. The network demonstrates that its prediction ability of real currents is slightly greater than the 38-constituent harmonic analysis with an $r^2$ of 0.978 and MAE of 0.048 ms$^{-1}$ in comparison to the harmonic predictions $r^2$ of 0.974 and MAE of 0.058 ms$^{-1}$ when compared to the ADCP data. Overall, this analysis proves the ADCPs worth in the measurement of small scale non-tidal and large one-off currents in the resource assessment. It also shows that this HF radar and ANN technique is a proficient tool for the long-term prediction of subsurface currents with few data records required, instead of prediction via harmonic analysis.

Despite the network's inability to predict currents at ADCP-E, harmonic composition was still carried out on the ADCP data and compared which helped identify an apparent higher magnitude ebb tide in comparison to flood at the site. This asymmetry is likely due to bathymetrical features as it is not present in the harmonic prediction, for example, the various shapes of a shelf seabed can cause the $M_2$ and $M_4$ generated tidal currents to combine [56], causing a stronger tide in one direction. The network was able to pick up on this asymmetry as it is an often-recurring feature. However, this resulted in the network predicting a strong asymmetry, sometimes up to 0.2 ms$^{-1}$ on every ebb tide, whereas the asymmetry was more varied in the measured data. A potential reason for this failing is the recurrent architecture of the network. It uses the happenings at the previous time steps for the prediction of the next time step [27], meaning that at the approach to peak ebb tide, the rate of decrease in velocity may have been such that the network did not expect the peak velocity to occur so soon, hence the overprediction. Despite its failing, it does show the network's ability to identify asymmetry at a site which is useful for tidal site characterisations as the deployment of tidal developments in asymmetric regions is likely to have a more pronounced impact on sediment dynamics [57,58]

## 5. Conclusions

This paper explored an alternative technique for tidal resource assessment using a land-based HF radar and an ANN to estimate subsurface currents from radar-measured surface currents.

After statistical testing of multiple network training methods and structures, the BR network with a single hidden neuron produced predictions most accurate to the measured data. This best network was able to generate both subsurface time series and vertical velocity profiles with >98% correlation to measured data (time series: r = 0.99, $r^2$ = 0.98, STD = 0.46 ms$^{-1}$, MAPE = 4.63%).

The ANN radar technique appears to be as powerful as a 38-constituent harmonic analysis for prediction at a single point, sometimes outperforming it with the ability to predict non-tidal currents. The high accuracy of the ANN method is produced from only a 2-week ADCP training dataset, closely matching the prediction of harmonic analysis which requires lengthy data acquisitions for high accuracy. The high accuracy of this method to measured data reduces the logistics required for long subsea measurements by reducing the deployment of offshore instruments from more than a year, to 2 weeks.

Applying the network at a site 10 km away proved that due to differences in the vertical structure of the water column between the training dataset and the new location, the network is not applicable for resource assessments over large areas where complex bathymetrical variation occurs.

The statistical tests showed that a neural network combined with constant HF radar surface current data is an effective method for the prediction of subsurface currents, in comparison to in situ-measured data and harmonic predictions, the results of which successfully produced data products and figures required for tidal resource assessments.

**Author Contributions:** Investigation, analysis, visualisation, writing, review, and editing, M.C.B.; conceptualisation, data resources, supervision, review, and editing, D.C.C. All authors have read and agreed to the published version of the manuscript.

**Funding:** The data used in this research was collected under support of the Natural Environment Research Council (Grant NE/J004219/1). The original analysis was performed for the lead author's MSc project and final preparation of the manuscript was supported by the European Union funded Marine-I (2nd phase) project (grant no. 05R18P02816).

**Data Availability Statement:** Data are soon planned to be available through Copernicus Marine Environment Monitoring Service (CMEMS). To request data, an email to the correspondence email address of this article will be answered.

**Conflicts of Interest:** The authors declare no conflict of interest.

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
