# Peer review of "Using Artificial Neural Networks for the Estimation of Subsurface Tidal Currents from High-Frequency Radar Surface Current Measurements"

_remotesensing, doi:10.3390/rs13193896_

Round 1
Reviewer 1 Report
This work provides a detailed presentation of high‐frequency radar processing scheme for the retrieval of surface currents using artificial neural networks. I think this is an interesting paper worth publishing with some modifications. Generally,it is well written, but please find below some remarks and suggestions which I think should be taken into account before the paper is suitable for publication.
(1) The introduction should be rewrite for logical formation, especially for the end of introduction.
(2)The references are relatively few in the last three years.
(3)Line 113, what is the mean of WECs, please give the full name in the paper.
Line 317, “this result in…”, what does “this” mean here?
(4)Conclusion should be revised to concentrate on the mainly contribute on this topic.
Author Response
The attached file represents our response to both reviewers.

Reviewer 2 Report
I have read carefully the paper and overall I believe it is an very interesting work, well written, worth publishing also with useful application on renewable energy issues. The paper focuses on the use of Artificial Neural Networks to predict current velocities, using data from high‐frequency radar. The results of the authors are highly correlated with measured values. In my view the manuscript is well written and organized, the results are supported by the data, the illustrations are well presented.
My main comments are the following:
- In my view, the abstract should note how this method would contribute to renewable energy issues.
- The introduction needs to address the aims of the paper.
- The methods section is rather long, I would propose to organize into sub-chapters
Overall, I my suggestion is that the paper should be accepted with minor revisions
You may find in attachment the annotated pdf with some minor corrections for the authors.

Author Response
Our response to both reviewers is uploaded here.

Round 2
Reviewer 1 Report
The manuscript can be published in this version.